# High Reynolds number investigations on the ability of the full scale e-Telltale sensor to detect flow separation on a wind turbine blade section

Antoine Soulier[1,2], Caroline Braud[2,3], Dimitri Voisin[1], and Frédéric Danbon[3]

[1]Mer Agitée, Port-la-Forêt, 29940 La Forêt-Fouesnant
[2]LHEEA lab. (CNRS/Centrale Nantes), 1, rue de la Noë, 44321 Nantes
[3]CSTB, 11 Rue Henri Picherit, 44300 Nantes

**Correspondence:** caroline.braud@ec-nantes.fr

**Abstract.** The complexity of the flow over a wind turbine blade makes its understanding and monitoring a challenging task, especially on operating wind turbines. The innovative e-Telltale sensor is developed for that purpose : detecting the flow separation on wind turbines blades. In this paper, high Reynolds number wind tunnel tests have been performed with different configurations of full scale e-Telltale sensors and wall pressure measurements on a wind turbine blade section. A comparison between the lift curve and the e-Telltale signal was used to evaluate the ability of the sensor to detect flow separation. Results show different interesting properties of the sensor response depending on its size, position along the chord and its fitting process that could be used in real applications.

## 1 Introduction

Increasing the life of wind turbines is one of the major areas of investigation faced by wind farm operators. A cause of premature aging often put forward is the accumulation of loads imposed by the strong shears upstream of the rotor due to an incorrect setting of the wind turbine or to the atmosphere in which it operates (Rezaeiha et al., 2017). In order to limit the influence of these disturbances on the wind turbine, modern pitch-regulated wind turbines are operational today. Sensors currently used are located on the wind turbine nacelle such as cup anemometers (Smaïli and Masson, 2004), on the wind turbine spinner (Pedersen et al., 2015). Upstream flow measurements from a LIDAR mounted nacelle are also under development (Scholbrock et al., 2013). This LIDAR system will predict flow perturbations (gust, misalignment . . . ) before a control action of the blade is performed. The control objective is to alleviate turbine blade lift fluctuations and resulting load fluctuations by an adequate adjustment of the blade incidence. However, measurements at the wind turbine nacelle do not sufficiently take into account the state of the flow on the aerodynamic surfaces (attached/separated, laminar/transitional or turbulent aerodynamic boundary layer . . . ) which is critical to decide a control action. Also, pitch control on very large blades is not fast enough to account for the small time scales from small turbulent structures generated in a wind turbine wake for instance (Chamorro et al., 2012), while they have a strong impact on blade loads (Bartholomay et al., 2018). Having local and robust aerodynamic sensors at the blade scale placed at key areas would be an important step for wind turbine monitoring and operation. Furthermore, they

could be used together with active devices to further decrease local spatio-temporal loads (Shaqarin et al., 2013; Jaunet and Braud, 2018). Swytink-Binnema and Johnson (2016) have demonstrated the possibility to detect aerodynamic flow separation using distributed tufts over the blade surface and a root-blade embedded camera. A simple and robust alternative of this sensor is the use of electronic-Telltale sensors. The system is composed of a silicone strip with a strain gauge at its base. When the silicone strip goes away from the surface, the displacement is measured by the strain gauge. The e-Telltale sensor is already used on sails of some boats and can be glued on the aerodynamic surface of wind turbine blades for retrofitting purposes. First tests of this innovative sensor were conducted at low Reynolds number using a down-scaled device (Soulier et al., 2021). The strip of the e-Telltale was demonstrated to follow the separation/reattachment dynamics similarly as much more accurate detection methods based on Time Resolved PIV measurements. The present paper extends the investigation on the ability of the e-Telltale sensor to detect the flow separation over airfoil profiles towards the use of a full scale device and high Reynolds number wind tunnel tests (the chord based Reynolds number is $Re_c = 8.85 \times 10^5$). The same 2D blade section was used here (NACA-$65_4$-421) and different parameters of the e-Telltale device were investigated, including its position, the strip length and its surface fitting process. The evaluation has been performed at different AoAs and through measurements of three lines of chord-wise pressure taps acquired synchronously with the strain gauge signal of the e-Telltale sensor.

The paper is divided in three main sections. Section 2 describes the experimental set-up, including the description of the wind tunnel facility, the blade manufacturing, the measurements used for the evaluation of e-Telltale sensor and the description of the different e-Telltale configurations. Section 3 describes the aerodynamics of the chosen blade profile, focusing on the flow separation phenomena. The last section present the results including the impact of the strip location, length and fitting process on the sensor signal.

## 2   Experimental setup

### 2.1   The wind tunnel

The measurements were performed at Nantes (France), in the NSA return wind tunnel of CSTB[1]. The test section is 20 m long with a cross-section of 4 m $\times$ 2 m (see figure 1). The turbulence intensity level in this test section is around 1 % and the operating speed of the wind tunnel is set to $U_\infty = 20$ ms$^{-1}$. At this free stream velocity, the Reynolds number of the flow based on the chord length ($c = 0.693$ m) is $8.85 \times 10^5$. The profile was set on a rotating table on its bottom, and guided on the ceiling using a bearing. Three pressure lines were used at different span locations, where measurements were performed, to check the flow bidimensionality in the area of interest (see section 2.3).

### 2.2   Blade manufacturing

The chosen 2D blade section, NACA $65_4$-421, was manufactured to be installed in the wind tunnel facility of CSTB. It was made of a skeleton coated with a supple composite. Details on the blade manufacturing can be found in figure 2. The shape

---

[1]http://www.cstb.fr/en/

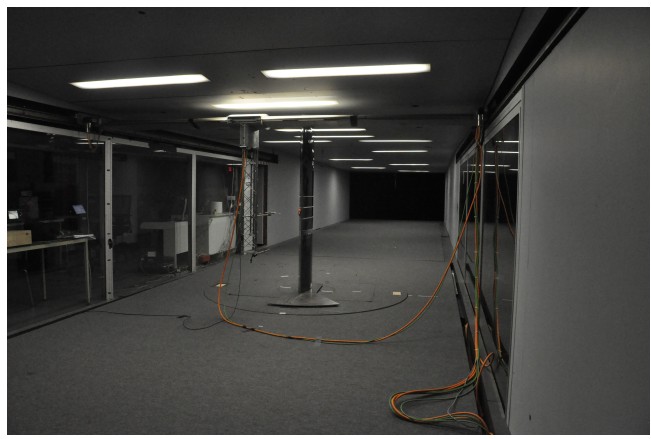

**Figure 1.** The NSA wind tunnel facility of CSTB with the 2D NACA $65_4$-421 blade section installed

of this airfoil profile, already used in previous studies (Sicot et al., 2008; Devinant et al., 2002), is also used in operation on stall-regulated wind turbines.

## 2.3 Measurements

The coordinates axis of the present study is $x$ in the streamwise direction, $y$ in the cross direction and $z$ in the spanwise direction perpendicular to the chord line. The origin is taken at the intersection between leading edge and the chord line, in the middle of the blade span (see figure 3). To get the lift coefficient $C_L$, three chord-wise lines of pressure taps were distributed around the profile using three 3D printed ribs equipped with 117 pressure sensors each. They were located in the middle of the profile at $z = 0$, and $z \pm 0.173\,c$ (see figure 3). Copper tubes of 0.8mm internal diameter were flush mounted using pressure tap holes in the 3D printed ribs. Vinyl tubes were then connected to transport the pressure towards ESP 32HD pressure sensors ranging from 0 to 2.5 kPa with a precision of $\pm\,0.03\,\%$ of the full scale. The cut-off frequency of the total system (tubes plus sensors) was 256 Hz. The signal was low-pass filtered at 256 Hz and acquired at 512 Hz. For each $AoA$, the pressure coefficient, $C_P$, was calculated and averaged on the duration on the measurement which was 2 minutes, then the lift coefficient $C_L$ was computed for each line of pressure taps by integration. It has been checked that the statistical convergence of $C_L$ is reached well before 1 minute. The lift coefficient presented in this study are the results of an average over the three lines of pressure.

### 2.4 E-telltale sensors

E-Telltale sensors are composed of a silicone strip with a strain gauge sensor at its base, so that it measures displacements of the strip away from the surface (see figure 4). The signal from the strain gauge sensor was amplified with a specialized amplifier (LTC6915). The signal is then filtered with a low-pass analog filter with a cut-off frequency of 160 Hz and a slope of $-20$ dB per decade. The filtered signal was then recorded with the same acquisition device than for the pressure measurements.

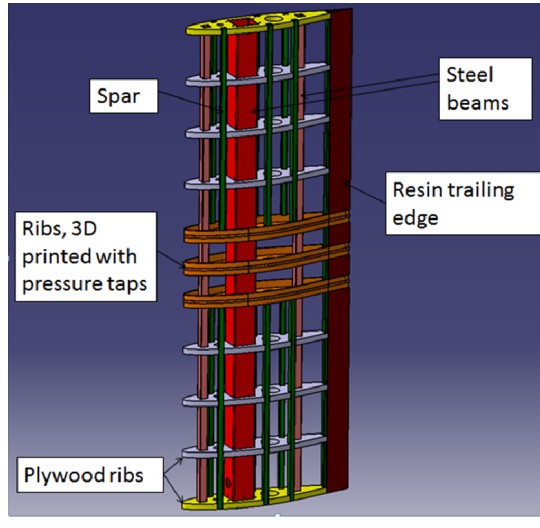

a)

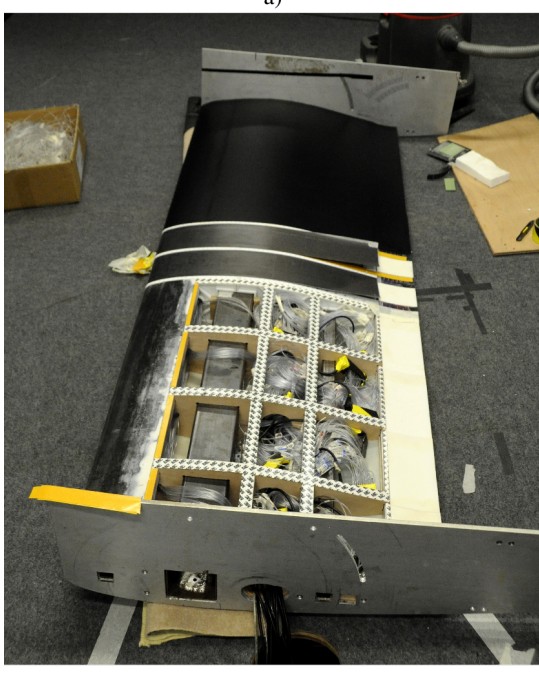

b)

**Figure 2.** The blade manufacturing. a) Elements of the skeleton: 2 steel beams ensure the rigidity, Plywoods ribs are ensuring the blade aerodynamic shape, 3D printed ribs are equipped with pressure taps, The trailing edge is made of resin. b) Picture of the blade with one of the GRP skins removed.

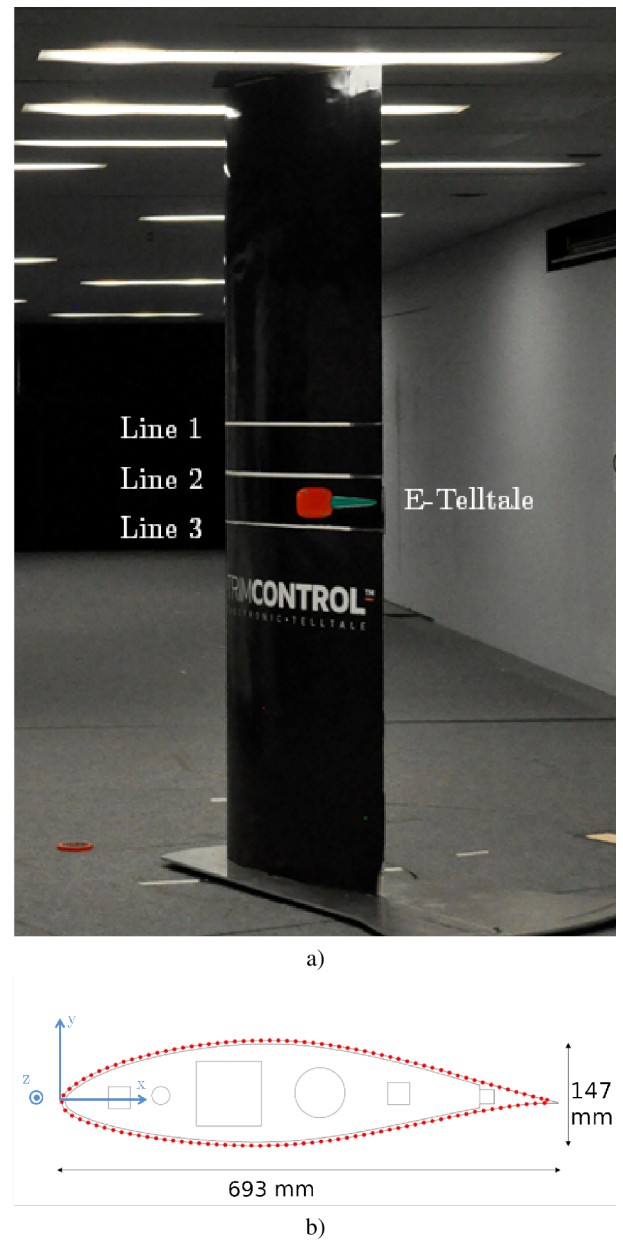

a)

b)

**Figure 3.** a) Positions of the chord-wise pressure lines along the span b) positions of pressure taps around the chord

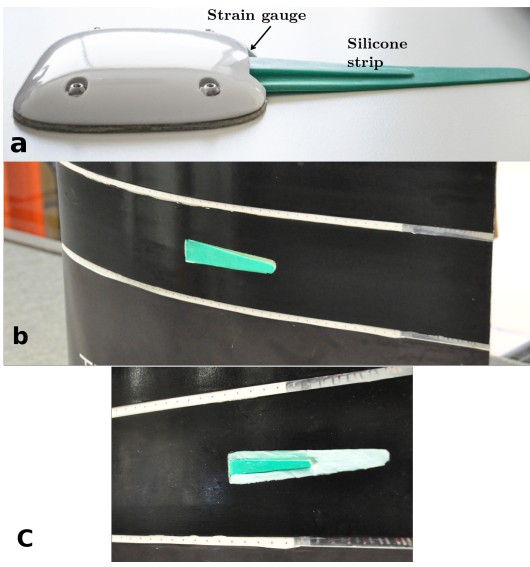

**Figure 4.** Picture of the innovative E-Telltale sensor (a : Shell and long strip cases of Table 1, b : no shell and long strip cases of Table 1, c TENSS case of Table 1) .

In the present study, which is a first high Reynolds number wind tunnel evaluation of this innovative e-Telltale sensor, we focuses on the ability of the e-Telltale sensors to detect mean load variations highlighted by the lift coefficient evolution versus the AoA and related to the flow separation over the suction side surface of the blade described in section 3. Different parameters
of the e-Telltale sensor were investigated and summarized in table 1. The first question that arises for such local sensors is where to place them on operating turbines. For this reduced problem (2D blade model), this question can be summarized as follows: which sensor positions are the best to detect the angles at which separation occurs ? Two locations will be targeted as this blade profile shape has two types of flow separation (see section 3 for more details): the flow separation at the trailing edge and the flow separation at the leading edge. Therefore, the end of the strip is first located at $95.6\ \%$ from the leading edge, the
Trailing edge or TE case, and at $31.8\ \%$ from the leading edge, the Leading-edge or LE case (see figure 5). The TE position is slightly modified for shorter strips so that the end of the strip is at $87.6\ \%$ from the leading edge rather than $95.6\ \%$ (see figure 5). Another question we target to answer in this study is the ability of this device to detect flow separation for two different surface fitting processes. The first e-Telltale is equipped with an aerodynamic shell that is glued on the surface, the Shell or S case. The second e-Telltale is integrated in the surface of the airfoil so that the surface of the airfoil is less perturbed, the
NoShell or NS case. The shell case is generally mounted on wind turbines already in operation, while the NoShell case could be part of the blade manufacturing process. As a first evaluation of this parameter, the length of strip is divided by almost two by keeping only the central part of the strip that is thicker (see figure 4 and 7). Between the two cases, the surface/thickness ratio is modified from 921 mm to 218 mm. The two cases will be referred later as the long (or L) and short (or S) strip.

| Name | Short Name | Location | Shape | Size ($L/c$) |
|---|---|---|---|---|
| TE-Shell-Long | TESL | Trailing edge (see figure 5) | with shell | 0.19 |
| TE-NoShell-Long | TENSL | Trailing edge (see figure 6) | without shell | 0.19 |
| LE-NoShell-Long | LENSL | Leading edge (see figure 6) | without shell | 0.19 |
| TE-NoShell-Short | TENSS | Trailing edge (see figure 7) | without shell | 0.11 |

**Table 1.** e-Telltale configurations (with $L$ the length of the silicon strip and $c$ the chord of the profile)

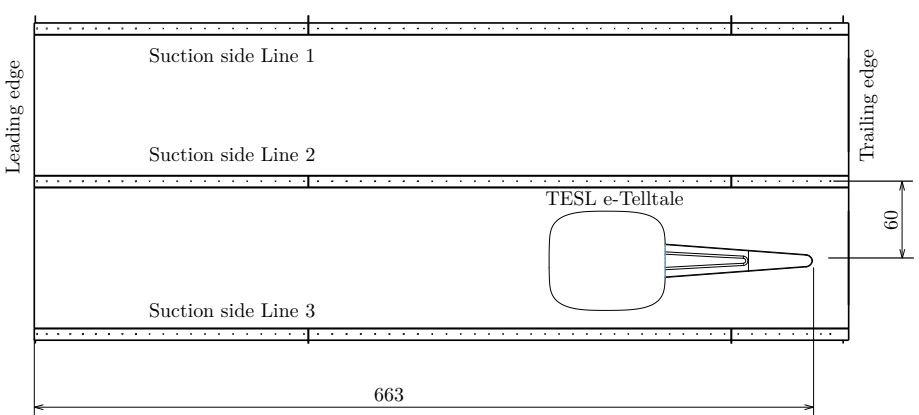

**Figure 5.** Position of the TESL e-Telltale

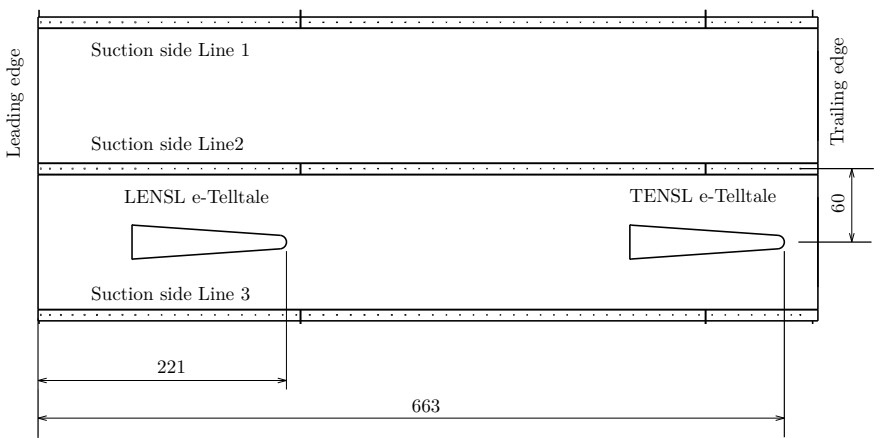

**Figure 6.** Positions of LENSL and TENSL e-Telltales

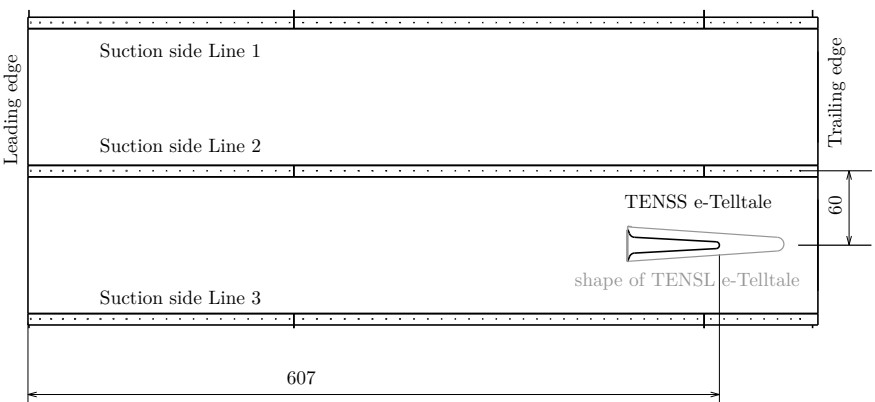

**Figure 7.** Position of TENSS e-Telltale

## 3   Blade aerodynamics

The NACA-$65_4$-421 profile has different slope modifications of the static lift coefficient corresponding to different states of flow on the suction side surface of the airfoil. This profile was already studied in the work of (Devinant et al., 2002) and a modified version (rounded trailing edge) of this NACA profile was used in the ANR (French national grant) project SMARTEOLE (Leroy, 2018; Braud and Guilmineau, 2016; Baleriola et al., 2018; Jaunet and Braud, 2018).

  Measurements of the present study were performed with a turbulence intensity of $1\,\%$, which may be at the origin of some
95 discrepancies in the exact values of the separations angles from the different studies on the same profile (Devinant et al., 2002) . However, similarly to previous studies, this profile shape presents different states of the flow depending on the AoA that have an impact on the shape of the lift curve and on the chord-wise pressure distribution. Figures 8 and 9 show lift coefficient and pressure coefficient of the average of the 3 lines of pressure taps, they are used for a rough description of the flow state with the AoA for this airfoil shape as there exist many scenarios according to Gault (1957):

– Until $AoA \simeq 6\,°$, the lift rises linearly with the AoA, the flow is attached to the surface of the profile as seen on the pressure distribution with no area of constant pressure on the suction side of the trailing edge.

  – Between $AoA \simeq 6\,°$ to $8\,°$, the flow is transitioning from the attached state to the separated state (i.e. zero pressure gradient), as can be seen on the zoom of figure 9, where the mean pressure coefficient near the trailing edge is progressively increasing towards a plateau from $AoA = 6\,°$.

– From $AoA \simeq 8\,°$ to $17°$, the flow separation (i.e. mean zero pressure gradient) can now be observed in average and move progressively towards the leading edge. This corresponds to a linear evolution of the lift with the AoA, with however a smaller slope than the previous flow state.

  – From $AoA \simeq 17\,°$ to $20\,°$, the evolution of the separation point towards the leading edge is faster and not linear, the flow is transitioning towards stall. On the pressure distributions at $AoA = 17\,°$ and $AoA = 20\,°$ a large plateau of constant

wall pressure can be seen on the suction side of the profile showing that the flow separation location reaches the first 30 % of the chord lenght.

- – Over $AoA \simeq 21\,°$ the separation point has reached the leading edge area, the flow is stalled and the wall pressure is almost constant on the whole suction side. The flow behaves like an asymmetric bluff body with shear layers on each side of the profile (from the leading edge and the trailing edge), a recirculating area in the close wake, and a wake behavior

  further downstream. Form the lift curve, the flow can be considered as a stalled flow.

These flow states correspond to a progressive displacement of the mean flow separation from the trailing edge to the leading edge, until the stall occurs, a typical scenario found for thick airfoil shapes by Gault (1957). It should be noted that, near the transitioning regions (around AoA 8° or near stall) the flow becomes 3D as highlighted by (Manolesos et al., 2014; Bak et al., 1999), leading to discrepancies that are still under investigations (Olsen et al., 2020). Therefore, for consistency, the pressure

measurements were conducted simultaneously with the e-Telltale sensor signal and they are presented with the e-Telltale signal. Also, an average of the two chordwise pressure lines that surround the e-Telltale sensor is used for the following plots.

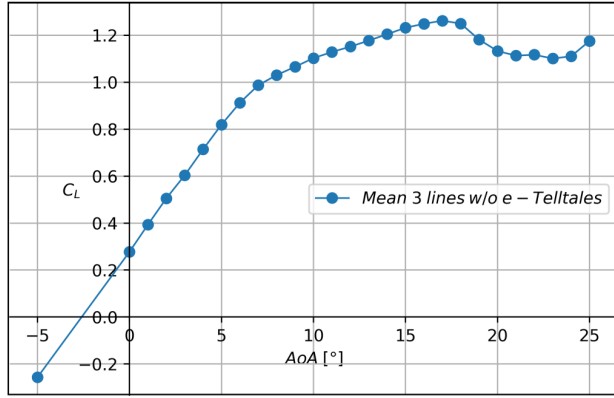

**Figure 8.** Lift coefficient mean of 3 lines

## 4   Results

During these experiments the attention was drawn to evaluate the ability of the e-Telltale to detect the slope changes on $C_L$ corresponding to the flow separation at the trailing edge ($AoA \simeq 8\,°$) and the leading edge flow separation, corresponding to

the stall angle ($AoA \simeq 21\,°$) as explained in section 3. Each modification of the lift slope corresponds to an evolution of the flow separation state, which ideally induces a different movement of the e-Telltale strip and thus a difference in the strain gauge signal of the e-Telltale device. In this section, the time averaged strain gauge signal and the standard deviation are computed for each AoA and presented together with the corresponding lift curve.

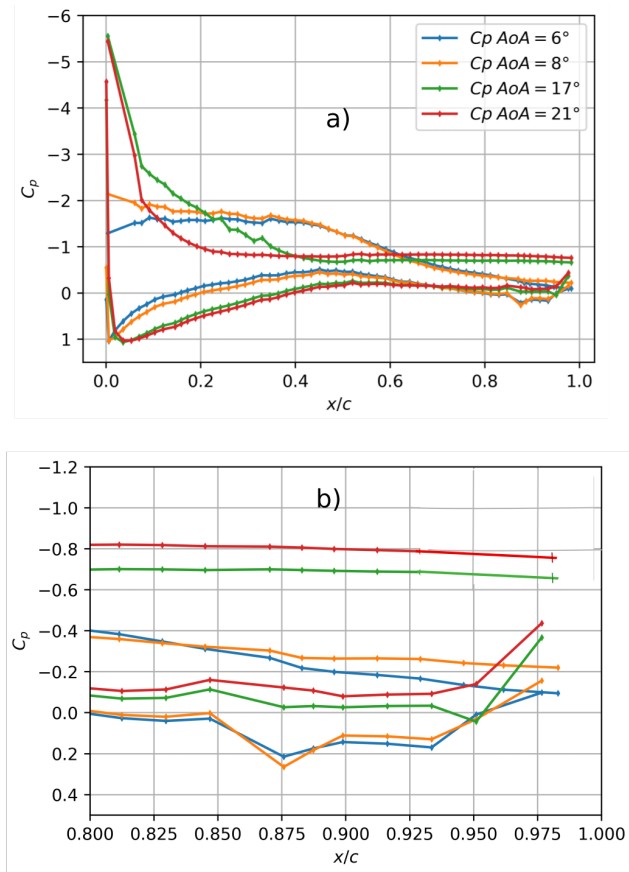

**Figure 9.** a) Pressure coefficient $C_P$ average on the 3 lines of pressure taps for $AoA = 6°, 8°, 17°, 21°$, b) zoom at the trailing edge

### 4.1 The reference case (TENSL)

In order to compare different parameters of the e-Telltale (as presented in the table 1) the TENSL configuration (near the trailing edge, no shell, long strip) was selected as the reference case. The figure 10 shows the evolution of the mean and the standard deviation of the signal of the e-Telltale corresponding to the TENSL case as function of $AoA$. The lift curve is plotted together to allow us to locate the state of the flow with the AoAs which are recalled in the figure. From -5° to 5°, the linear evolution of the lift indicates an attached flow state. For this $AoA$ range, both the mean and the standard deviation of the signal of the e-Telltale are near 0. From 5° to 8°, corresponding to the flow separation appearance at the trailing edge, both the mean and the standard deviation rise up with the $AoA$. Then from 8° to 10° they decrease to values near 0. It should be noticed that this behaviour is not present when the e-Telltale is located near the leading edge (LENSL, figure 12), clearly indicating that this bump from 5° to 10° is associated to the location of the e-Telltale sensor in the trailing edge flow separation area. This suggests that, in that region, when the AOA is increasing from 5° to 8°, the flow is departing from the wall causing movement of the e-TellTale strip outward the wall, thus increasing its signal. The separated flow should naturally induces a mean shear

area with associated turbulent structures. This turbulent separated shear layer is supposedly fluctuating from the wall to the separated shear area, causing fluctuations of the e-Telltale strip (that was observed), increasing the RMS value of the e-Telltale sensor. Above $8\,°$, the separated shear layer is probably moving to far from the wall, getting out-of-reach of the e-TellTale strip length, which in turn reduces the mean and RMS value of the e-TellTale signal. More spatio-temporal informations are

145 needed to confirm this scenario. Also, it should be noted that this scenario do not take into account potential fluid-structure interactions with the e-TellTale strip. This is however interesting to notice that the e-Telltale has a particular sensitivity to the transition flow state from the fully attached flow to the trailing edge flow separation. The advantage of this observation is to make this innovative sensor appropriated to predict the trailing edge flow separation. At $AoA > 10\,°$, both the mean and the standard deviation of the e-Telltale signal increases again until $AoA = 18°$. Contrary to the $AoA$ range corresponding to the

150 appearance of the trailing edge flow separation, between $AoA = 5\,°$ to $AoA = 8\,°$, the evolution is not linear. A progressive saturation of the signal appears slowly. This may be explained by the displacement of the separation point and its associated shear layer, that are moving further away from the sensor with the increase of the $AoA$. Between $AoA = 18°$ and $AoA = 21°$ the e-Telltale signal is marked by a sudden rise in two steps. A first moderate step between $18\,°$ and $20\,°$ and a sudden rise between $20\,°$ to $21\,°$. This can be explained by the strip that is flipping towards the leading edge due to the strength of the

155 reverse flow after $20\,°$, in good agreement with low Reynolds number experiments performed by Soulier et al (Soulier et al., 2021). After the stall angle, $AoA = 21°$, the mean signal reaches its maximum value while the standard deviation has doubled. This indicates that the strip is not only flipping in the reverse flow in average, but it is also fluctuating in the flow with larger oscillation than when the stall has occurred, again in good agreement with low Reynolds number observations (Soulier et al., 2021). The significant rise which occurs at the stall angle allows a clear detection of the stall angle with the e-Telltale sensor,

even though not ahead its appearance. After this $AoA$ of $21\,°$, the stalled flow or the bluff-body flow is settled, both the mean and the standard deviation of the e-Telltale signal are linearly decreasing, probably due to the deportation of the shear layer further away from the location of the e-Telltale.

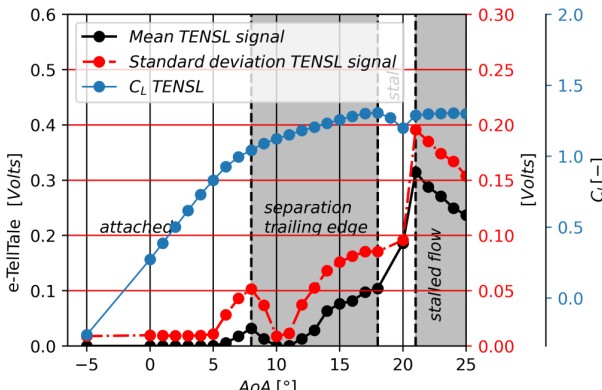

**Figure 10.** Mean and Standard deviation of the e-Telltale TENSL signal and lift $C_L$ as function of $AoA$

## 4.2 Influence of the strip stiffness or length (TENSS case)

Compared to the previous case (section 4.1), the TENSS case presents a shorter strip length with a reduction of 58% (see table 1). The TENSS case has also a surface-thickness ratio reduction of $77\,\%$ with $A/T$ varying from 921 mm to 218 mm ($A$ is the area of the strip and $T$ is the thickness of the strip), which parameter is inversely proportional to the stiffness of the strip. The short strip is therefore much stiffer than the long one.

The e-Telltale is located at the same position of the reference TENSL case (the root of the strip at around $77\,\%$ of the chord). For this test, the same device is used but the strip has been cut, ensuring that the strain gauge and the position of the strip are identical to the reference case. The figures 11a and 11b show respectively the mean and standard deviation of the TENSS case signal versus the AoA, plotted together with the reference case. It clearly shows that there is almost no information about the stall with the shorter strip case (i.e. for $AoA > 19\,°$), only a slight increase of the standard deviation value with a peak around $21\,°$ is observed (see figure 11b). On the contrary, for AoAs corresponding to the appearance of the flow separation at the trailing edge, from $AoA = 5\,°$ to $AoA = 8\,°$, the mean signal is 1.5 higher than the reference case (longer strip case). Also, the mean and standard deviation of the signal suddenly drops at $AoA = 9\,°$. Having a higher sensitivity of the strain gauge signal for the shorter strip case which is stiffer, highlights that the length of the strip is the most relevant parameter for the detection of trailing edge separation phenomena of this profile. For the stall phenomena it seems that the reversed flow close to the wall is not the relevant phenomena at the origin of the displacement of the strip, but rather the separated shear layer and its distance to the wall. However, further spatio-temporal explorations are needed to investigate this point. For further understanding of full scale blade aerodynamics, the high sensitivity of the short sensor case to the appearance of the trailing edge separation and not the stall, makes it really interesting to discriminate them.

## 4.3 Influence of the position of the e-Telltale (LENSL)

The influence of the position of the e-Telltale is discussed in this section. The e-Telltale located near the leading edge in the area of high favorable pressure gradient, the LENSL case, is compared to the reference case, at the trailing edge (see figure 6) in the figure 12. When located at the leading edge, the e-Telltale is clearly unable to detect the AoA corresponding to the trailing edge flow separation, $AoA = 8\,°$. On the contrary, both the mean and the standard deviation of the signal starts to rise just after the second linear part of the lift, $AoA = 18\,°$, with a stronger slope than when the e-Telltale is located at the trailing edge. This should be attributed to the separation location, where the zero mean pressure gradient starts, that is reaching the strip region, between $19\,\%$ to $31\,\%$ of the chord length, for angles of attack higher than $18\,°$. It leads to a higher level of the signal, more than doubled for the mean signal, and around $50\,\%$ higher for the standard deviation. Also, the peak of the signal occurs at higher AoAs than when the sensor is located at the trailing edge. All these observations are pretty obvious as they are in good agreement with the fact that the stall phenomena is originated at the leading edge, with the separated shear layer that remains, for the sensor located near the leading edge, close to the wall and thus close to the sensor, even at high AoA.

Due to its strong signal value, this configuration is well suited to detect the stall phenomena compared to other e-Telltale cases.

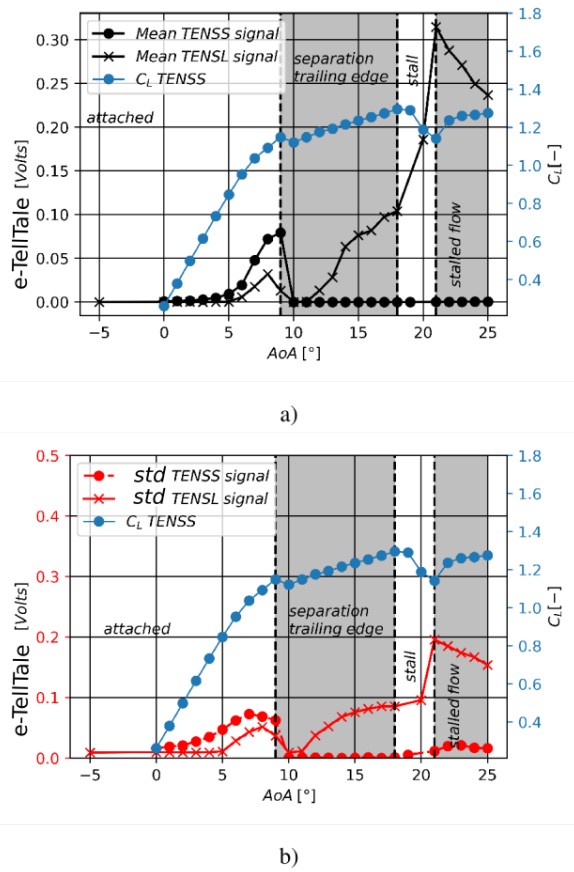

**Figure 11.** Comparison of the mean a) and the standard deviation b) of the signal of the TENSL (reference case) and TENSS e-Telltales as function of $AoA$

## 4.4 Influence of the shell (TESL case)

The figure 13a and 13b show respectively the mean and standard deviation of the TESL case versus the AoA, compared with the reference case. The magnitude of both, the mean and standard deviation is larger considering both the trailing edge separation and the leading edge flow separation angles. However, both phenomena are detected at higher $AoA$, $AoA = 9°$ and $AoA \simeq 22 - 23°$ respectively, when compared with the sensor integrated in the aerodynamic surface. We cannot explain the measured differences between the two configurations with the available measurements. Too many parameters are involved here to conclude. It should however be noticed that the strip of the e-Telltale in the integrated case (i.e. TENSL case) was observed to have difficulties to initiate a movement. This may be attributed to a complex interaction between the boundary layer with the cavity in which the sensor is integrated.

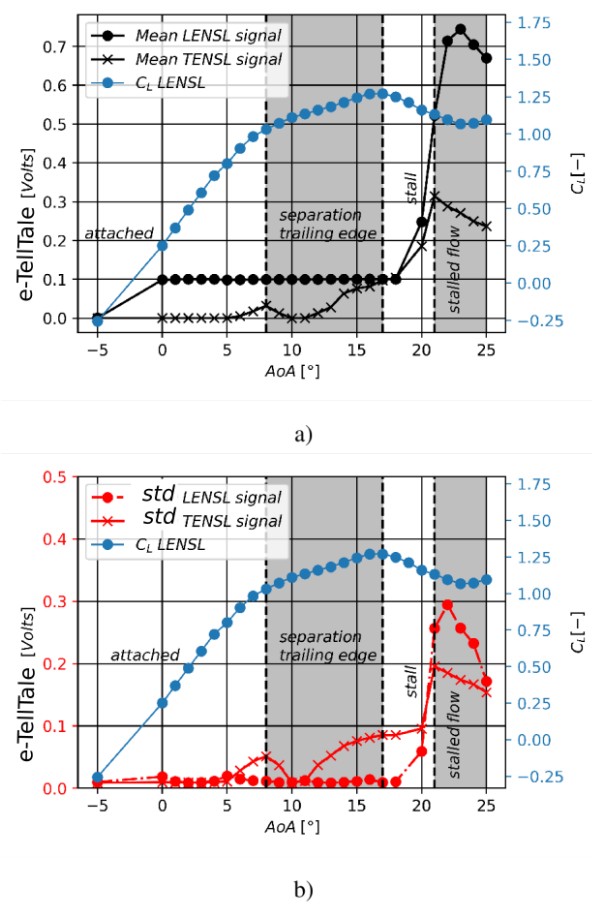

a)

b)

**Figure 12.** Comparison of the mean a) and the standard deviation b) of the signal of the LENSL and TENSL (reference case) e-Telltales as function of $AoA$

## 5   Conclusion

In order to evaluate the ability of an innovative aerodynamic sensor, the e-Telltale, to detect the flow separation on wind turbine blades, high Reynolds number wind tunnel tests ($Re_c = 8.85 \times 10^5$) were performed on a 2D blade section using a full scale e-Telltale sensor. This innovative sensor is made of a strip with a strain gauge at its base, detecting displacement of the strip away from the aerodynamic surface. These tests highlight the impact of different parameters on the signal of the e-Telltale, and also the possible different use of the sensor in agreement with these findings. First, the e-Telltale with a long strip located at the trailing edge is able detect both the trailing flow separation angle and the stall angle. This e-Telltale detects only the stall angle when located at the leading edge, with however a higher amplitude in the mean and standard deviation of the sensor's signal. For a sensor located at the trailing edge, the strip can be shortened to only detect the trailing edge separation angle with a higher sensitivity, this configuration being however ineffective to detect the stall angle. The limit size of the strip has

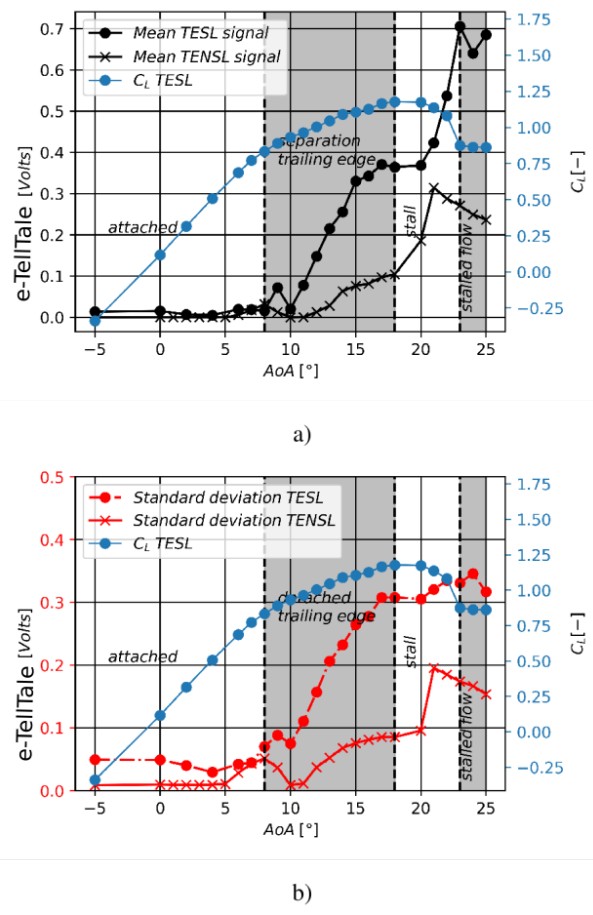

**Figure 13.** Comparison of the mean a) and the standard deviation b) of the signal of the TESL and TENSL e-Telltales as function of $AoA$

not been identified yet but a relative length of $L_{TENSL}/c = 0.11$ (associated with a surface to thickness ratio of $218$ mm) is short enough for that purpose while a length of $L_{TENSS}/c = 0.19$ (associated with a surface to thickness ratio of $921$ mm) is too long. When the e-Telltale is not integrated to the aerodynamic surface, which may concern all the wind turbines already in operation, the same conclusions can be drawn: this sensor is able to detect both the trailing edge flow separation angle and the stall angle with however a slight delay in the detection of the trailing edge separation regarding the lift curve used as the

reference. This delay in the detection of flow separation by the e-Telltale have been addressed at a smaller scale in (Soulier et al., 2021). However some future measurement are needed at full scale to complete the study of the delay. At last, the sensor might have some limitations when dealing with thin airfoils that have a strong stall just after the maximum lift value, with a very fast displacement of the flow separation from the trailing-edge to leading-edge. In that case the dynamic response of the sensor is of major importance and the best location might be rather near the leading-edge. Indeed, the sensitivity of the sensor

to the separated shear layer when it is located at the trailing edge may rapidly disapear.

*Author contributions.* This work was performed during the PhD of AS under the supervision of CB, DV and FD. CB provided her scientific expertise in turbulent shear flows and wind turbine blade aerodynamics, DV shared his expertise on the innovative e-Telltale sensor, FD supervised the wind tunnel experiments at CSTB.

*Competing interests.* No conflict of interest.

*Acknowledgements.* Authors would like to thanks the technical staff at CSTB who carried experiments in the NSA wind tunnel.

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
