# Peer review of "High Reynolds number investigations on the ability of the full scale e-Telltale sensor to detect flow separation on a wind turbine blade section"

_Wind Energy Science, 2021_

## Referee Comment (RC2)

**General Comments**

The manuscript addresses a very ineresting topic of relevance for the wind energy community by investigating the ability to detect flow separation with a novel mechincal sensor on the blade surface. In wind tunnel experiments at Re=885.000, different sensor configuration and the sensor placements along the chord are considered in order to gather information about trailing edge separation and leading edge separation. The general approach of comparing the sensor output signal and its standard deviation as a function of the profile AoA constitutes a reasonable measure for the usability and it is methodically sound. The presented results show the capability of the novel sensor to provide information about the onset of stall, which in some cases is limited due to the position of the sensor.
However, some of the conclusions stated are not clearly supported by the presented and discussed data.

I recommend this paper for publication after a revisions regarding the minor shortcomings addressed in the following.

**Specific Comments**

- In section 2.3, line 64, the authors mention, that the measuring time was either 1 or 2 minutes. An explanation for the two different measuring times should be given and the approach how to calculate comparable standard deviations from different measuring times should be made clear to the readers.

- On page 6, fig.4, a picture of the device with shell is presented. Since most of the results focus on the no-shell version, a picture of this setup shoud be included to provide the reader an improved understanding.

- On page 6, lines 92ff, the authors mention, that a version of the used blade profil with modified trailing edge has been used in other research. The relevance of this information is unclear and it should either be explained or ommitted.

- On page 8, line 102, the authors claim that the flow is transitioning from separated state to attached state between AoAs of 6° to 8°. With increasing AoA the transition should be towards flow separation.

- On page 8, line 105, the authors mention, that flow separation moves progressively twowards the LE up to an AoA of 18°. The decrease in $C_L$ ist already starting at 18° (fig. 8), though.

- The lift coefficients derived form the pressure taps in fig.8 deviate from the lift curves for the two cases with and without the mounted device in fig 9. Figure 9 shows a later transition to a lower slope in the linear region (6° vs. 7°) and also shows a higher maximum $C_L$ (<1.2 vs. >1.2). This leaves the readers with some questions about the reproducibility and reliabilty of the results. The authors should discuss these deviations and give an explanation. Also, since the device is mounted between two lines of

pressure taps, the comparison of lift curves for individual lines of pressure taps insetead of just averaged data would add value.

- In section 4.3 on page 11 the authors discuss the performace of the sensor when applied close to the leading edge. The finding of no detected separation for low AoAs should be expected, since the sensor is mounted in the region of the blade profile, where the flow is still attached. The finding seems obvious and this should be mentioned. The same holds for the finding, that separation close to the LE is detected once the AoA increases.

- The description of the impact of the shell in section 4.4 is very brief and superficial. From fig. 13 it is clearly visible, that the shell impacts the flow and thus the signal of the sensor. It can be expected, that the shell impact results in higher standard deviations for low AoAs. However, it is not clear why the signal is higher for fully separated flow in high AoA cases. No shell effect should be expected once the shell is located in the fully separated region. The authors unfortunately do not address the visible effects besides mentioning higher signal and standard deviation values for the shell case. A hypothesis and possible explanation of the effects would be helpful here. Unless this is addressed, this section provides no meaningful value to the paper and should be omitted.

- In the final sentence of the conclusion (p.13, line 202f), the authors claim, that the sensor with shell is also capable of detecting the TE separation angle and stall angle. This conlcusion is not fully supported by the plots in fig. 13 and the brief mention in section 4.4. While the local maximum of the standard deviation and the first increase of the signal value seem to be an indicator for TE separation, these increases occur with a slight delay compared to the $C_L$ curve used as a reference. The authors should be more precise in distinguishing the results for the different setups.

**Technical Comments**

The labeling of the plots in figure 10 to 13 lacks a proper label for the y-axes as only the unit [Volts] or no label (right y-axis) is given. Meaningfull labels should be added
The naming and abbreviations of the angle of attack varies between angle of attack, angle of ancidence, AoA, AOA. This should be check and modified for constistency.
The spelling and grammar is spotty in some sections. It is recommended to have thourough check of the language, preferably by a native speaker.

p.2, line 32:
Colloquial writing: [...]high Reynolds number wind tunnel tests.

p.3, line 58:
End of sentecne: "." missing.

p.3, line 62:
Unit: kPa.

p.3, line 63:
Unit: remove space in Hz.

p.8, line 101:
Additional ",".

p.8, line 102:
Betwenn -> Between.
Transitionning -> Transitioning.

p.8, line 103 and other instances throughout:
apparition -> appearance.

p.8, line 105:
Betwenn -> Between.
Transitionning -> Transitioning.

p.9, line 119:
[...] have not any [...] -> do not have an / have no

p.9, line 121:
slopes -> slope

p.9, line 123:
explain -> explained

p.9, line 124:
[...] of e-Telltale [...] -> [...] of the e-Telltale [...]

p.9, line 125:
average -> averaged

p.9, line 131:
[...] angle of incidence [...] -> [...] angles of incidence [...]

p.11, line 155:
[...] more stiff [...] -> [...] stiffer [...]

p.11, line 185f:
AoA vs. AOA: one abbreviation should be used consistenly during the entire paper

p.12, line 195:
e-TellTale -> e-Telltale

p.13, line 202:
[...] concerns [...] -> [...] concern [...]

---

## Referee Comment (RC3)

**Comment on "High Reynolds investigations on the ability of the full scale e-TellTale sensor to detect flow separation on a wind turbine blade section" by Antoine Soulier et al.**

**General comments**

In this manuscript, the authors intend to show the ability of the full scale e-TellTale sensor to detect the separation state of the flow on turbine blades.

The wind tunnel experiment of 2D NACA $65_4$-421 with 0.693m in chord length was conducted at Reynolds number of $8.85 \times 10^5$ with pressure measurement. Several configurations of e-TellTale sensor are tested on the surface and the signal was measured for each configuration. The result shows that the sensors can detect the separation states from both leading edge and trailing edge. It is also found that its sensitivity for different states are depend on its configuration.

This work presents an important evaluation of the innovative device for the progress of the sophisticated turbine control including active flow control technologies on the blade.

I strongly recommend this paper for publication with however revised to raise the reliability of the work. I hope the following comments help the authors for revision.

**Specific comments**

1. The authors failed to convince readers the explanation in Section 3 because there are no data. The authors should show the pressure distribution measured in this experiment to show the flow separation state for each slope of Cl.
2. The inconsistency between Cl in Figure 8 and 9 is also confusing for the readers to believe the reliability of this manuscript.
3. Cl for Figure 10-12 seems to be the same data. The authors should show measured lift curve for each experiment together with each strain gauge signal as explained in l.126 to show the ability of the sensor for the detection of the slope of Cl for each configuration. Otherwise, the authors should describe that the slope for all configurations are the same.
4. The authors should explain some features of the following Figures to make them reliable. For Figure 12, the authors should explain the reason why the Mean LENSL signals are kept 0.1 from 0 to 18 degrees of AOAs. For Figure 13, the authors should explain the possibility of interaction of the Shell of the sensor on the standard deviation of TENSS signal at low AOAs.

**Technical corrections**

Errors or strange phrases should be checked as follows. I strongly recommend the review by a native English speaker.
- Title: High Reynolds investigation → High Reynolds number investigation
- P.1, l.3: high Reynolds → high Reynolds number
- P.1, l.10: strong shears upstream of the rotor due to a malfunction of the wind turbine: check the phrases.

- P.1, l.11: In order to limit the influence of these disturbances on the wind turbine, modern pitch-regulated wind turbines are operational today. : Is this means the IPC system?
- P.1, l.15: The former → The latter?
- P.1, l.17: rotation → adjustment?
- P.2, l.25: silicon → silicone
- P.2, l.32: high Reynolds → high Reynolds number
- P.2, l.33: 8.85 → 8.85 x
- P.3, l.56: framework → coordinate axis?
- P.3, l.56: y i → y in
- P.3, l.68: silicon → silicone
- P.3, l.72: high Reynolds → high Reynolds number
- P.3, l.73: lift coefficient → lift coefficent variations?
- P.3, l.73: angle of incidence → AOA? Use the same words throughout the manuscript.
- P.4, Figure 4.: Also show a picture of the TENSS type for the readers understanding.
- P.4, l.80: What is "this last position"?
- P.4, l.86: "the more rigid the strip is, the less signal is transmitted": Is this a result of the experiment?
- P.4, l.87: "by keeping only the central part of the strip that is thicker": Add a picture with Figure.4 to understand this phrase.
- P.4, l.91: slope modifications → slope?
- P.5, l.96: "which may be at the origin of some discrepancies in the exact values of the separations angles.": discrepancies from what?
- P.5, l.98: separation → separation point?
- P.5, l.102: separated ←→ attached?
- P.5, l.102: separated ←→ attached?
- P.5, l.108: not always: Did you repeat the test?
- P.5, Figure 8: Make clear the border line of the figure.
- P.6, Figure 9: Make clear the border line of the figure.
- P.6, l.115: pressure distributions: How do we know the pressure distribution from the lines?
- P.6, l.118: There is not any significant difference: There are indeed differences.
- P.6, l.121: slope change → slope?
- P.10, l.135: show the region of the transition flow state in Figure 10.
- P.10, l.147: low Reynolds → low Reynolds number
- P.10, Figure 10: Add proper labels and units on each axis.
- P.11, l.154: Define A and T.
- P.11, l.170: Refer Figure 12 in this subsection.
- P.11, l.171: trailing → leading
- P.12, Figure 11: Add proper labels and units on each axis.
- P.13, Figure 12: Add proper labels and units on each axis.
- P.12, Figure 12: Add proper labels and units on each axis.
- P.14, l.205: shared is → shared
- P.15, l 213: Bartholomay2018: Correct the title.
- P.15, l 233: Shaquarin2013: Correct the authors.
- P.15, l 241: Swytink2016: Correct the title.

---

## Author Comment (AC1)

**Response to reviewers, paper wes-2021-4**

Antoine Soulier, Caroline Braud, Dimitri Voisin, and Danbon Frédéric

**1  Referee # 1**

**General comments**

In this manuscript, the authors intend to show the ability of the full scale e-Telltale sensor to detect the separation state of the flow on turbine blades.

The wind tunnel experiment of 2D NACA $65_4-421$ with $0.693m$ in chord length was conducted at Reynolds number of $8.85.10^5$ with pressure measurement. Several configurations of e-Telltale sensor are tested on the surface and the signal was measured for each configuration. The result shows that the sensors are able to detect the separation states from both leading edge and trailing edge. It is also found that its sensitivity for different states are depend on its configuration.

This work presents an important evaluation of the innovative device for the progress of the sophisticated turbine control including active flow control technologies on the blade.

I strongly recommend this paper for publication with however revised to raise the reliability of the work. I hope the following comments help the authors for their revision.

**Specific comments**

**Q1 :** The authors failed to convince readers the above explanation in Section 3 because there are no data. The authors should show the pressure distribution taken in this experiment to show the flow separation state for each slope of Cl.
The mean pressure distribution corresponding to the Cl curve has been added as suggested. Descriptions related to these quantities have been updated accordingly and shown in red in the final document. It should be emphasized that this description is only there to give a rough description of the stall scenario as it highly depends on the airfoil shape as found by the early study of Gault (1957).

**Q2 :** The inconsistency between Cls in Figure 8, 9 is also confusing for the readers to believe the reliability of this manuscript.

Thank you for noticing. The inconsistency comes from two main problems. The first confusion is due to an AoA correction we unfortunately did not reported in the figures of the original manuscript. The other inconsistency comes from the use of two distinct wind tunnel campaigns in the manuscript. As shown in figure 2, one campaign is relatively well 2D, the other one (figure 1 below) exhibit more 3D effects in the spanwise direction. As recalled in the final article, as soon as the flow separates, the flow becomes 3D (see e.g. Manolesos et al. (2014)), leading to lift discrepancies, which origin are still under investigation (see e.g. Olsen et al. (2020)). In the present study, it is though that changing the e-Telltale configuration modify slightly the blade shape and thus the lift level. However, understanding these 3D effects is out of the scope of the present paper. In order to be consistent in the description of the e-Telltale sensor's response, the simultaneous and local measure of the CL curve (via the closest pressure measurements) is now shown instead of the spanwise averaged CL value. However, for a rough presentation of stalled scenario of the present blade shape (see e.g. Gault (1957) for example of stall scenario with the Reynolds number and the airfoil geometry), the averaged CL value for the two campaign test is presented in section 3.

[Figure]

**Figure 1.** Lift coefficient of 3 lines with out e-Telltales, selected wind tunnel test campaign

**Q3 :** Cl's for Figure 10-12 seems to use the same data. The authors should show measured Cl slope for each experiment as explained in l.126 to show the ability of the sensor to detect the slope for each configuration. Otherwise, the authors should show the slope for each configuration are the same in another Figure.
The plot have been changed, for each e-Telltale configuration, the corresponding Cl is now presented

**Q4 :** The authors should explain some features in Figures to make them reliable. For Figure 12, the authors should explain the reason why the Mean LENSL signal is 0.1 for low AoAs
The offset of the sensor signal is really sensitive to its environment as the way it is mounted. This is why the analysis focuses on the changes of slope and the main features of the curve.

[Figure]

**Figure 2.** Lift coefficient of 3 lines with out e-Telltales, former version, non-selected wind tunnel test campaign

**Q5 :** For Figure 13, the authors should explain the possibility of interaction of the Shell of the sensor for the Standard deviation of TENSS signal at low AoAs

The interaction of the shell with the sensor is not raised in the article because no data could either confirm or deny the effects of the interaction. However what have been observed at least visually is that the behavior of the silicon strip is significantly different between a shell and a no shell e-Telltale. To go further into this analysis, additional dedicated measurements are needed such as 3D drag measurements or/and spatio-temporal measure of the flow field together with a measure of the strip movement, similarly as what has been performed at a lower scale experiment (see Soulier et al. (2021a)). This is however out of the scope of the present paper.

**Technical corrections** The technical corrections have been treated directly in the manuscript

**2  Referee #2**

**Q6 :** In section 2.3, line 64, the authors mention, that the measuring time was either 1 or 2 minutes. An explanation for the two different measuring times should be given and the approach how to calculate comparable standard deviations from different measuring times should be made clear to the readers.

As explained in answer Q2 from the first reviewer, results from several wind tunnel test sessions were used in the former version of the manuscript. For this updated version, data from a unique wind tunnel campaign are used, all the signals were acquired with a duration of 2 minutes. It has been updated in the manuscript.

**Q7 :** On page 6, fig.4, a picture of the device with shell is presented. Since most of the results focus on the no-shell version, a picture of this setup should be included to provide the reader an improved understanding.

The figure has been modified with an additional picture of the no-shell case.

**Q7 :** On page 6, lines 92ff, the authors mention, that a version of the used blade profile with modified trailing edge has been used in other research. The relevance of this information is unclear and it should either be explained or ommitted.

It has no specific relevance and has been omitted as suggested by the reviewer.

**Q8 :** On page 8, line 102, the authors claim that the flow is transitioning from separated state to attached state between AoAs of 6 ◦ to 8 ◦ . With increasing AoA the transition should be towards flow separation.

Thank you for noticing. This error has been corrected in the manuscript

**Q9 :** On page 8, line 105, the authors mention, that flow separation moves progressively towards the LE up to an AoA of 18 ◦ . The decrease in CL is already starting at 18 ◦ (fig. 8), though.

We agree with the reviewer, the AoA of 18° is, in this case, not the good one, the stall begin at the AoA of 17° corresponding, to the maximum lift. This have been changed in the manuscript.

**Q10 :** The lift coefficients derived form the pressure taps in fig.8 deviate from the lift curves for the two cases with and without the mounted device in fig 9. Figure 9 shows a later transition to a lower slope in the linear region (6 ◦ vs. 7 ◦ ) and also shows a higher maximum C L (<1.2 vs. >1.2). This leaves the readers with some questions about the reproducibility and reliabilty of the results. The authors should discuss these deviations and give an explanation. Also, since the device is mounted between two lines of 1pressure taps, the comparison of lift curves for individual lines of pressure taps instead of just averaged data would add value.

Please refer to our answer of Q2 from the referee #1

**Q11 :** In section 4.3 on page 11 the authors discuss the performance of the sensor when applied close to the leading edge. The finding of no detected separation for low AoAs should be expected, since the sensor is mounted in the region of the blade profile, where the flow is still attached. The finding seems obvious and this should be mentioned. The same holds for the finding, that separation close to the LE is detected once the AoA increases.

Thank you the advice, it has been added in the final version.

**Q12 :** The description of the impact of the shell in section 4.4 is very brief and superficial. From fig. 13 it is clearly visible, that the shell impacts the flow and thus the signal of the sensor. It can be expected, that the shell impact results in higher standard deviations for low AoAs. However, it is not clear why the signal is higher for fully separated flow in high AoA cases. No shell effect should be expected once the shell is located in the fully separated region. The authors unfortunately do not address the visible effects besides mentioning higher signal and standard deviation values for the shell case. A hypothesis

and possible explanation of the effects would be helpful here. Unless this is addressed, this section provides no meaningful value to the paper and should be omitted.

It is true that we cannot explain the differences between the two configurations with the available measurements. However, it is still interesting to point-out that the differences, that cannot be seen in the lift level (figure 10), are so strong in the e-Telltale signal. Too many parameters are involved here to conclude. It should however be noticed that the strip of the e-Telltale in the integrated case was observed to have difficulties to initiate a movement. This may be attributed to a complex interaction between the boundary layer with the cavity in which the sensor is integrated. More dedicated measurements are needed for further understanding. This hypothesis is addressed in the final document

**Q13 :** In the final sentence of the conclusion (p.13, line 202f), the authors claim, that the sensor with shell is also capable of detecting the TE separation angle and stall angle. This conclusion is not fully supported by the plots in fig. 13 and the brief mention in section 4.4. While the local maximum of the standard deviation and the first increase of the signal value seem to be an indicator for TE separation, these increases occur with a slight delay compared to the CL curve used as a reference. The authors should be more precise in distinguishing the results for the different setups.

The reviewer raised an interesting question that we unfortunately cannot answer with the available measurements. Indeed, if the pressure distribution is able to tell us when the flow separates, we do not have any information on the motion of the e-Telltale strip within the flow field, which prescribe any understanding of this sensor response delay. This is something we addressed at lower scale for the second peak of the e-Telltale signal (stall phenomena), with analysis of the strip motion using TRPIV measurements (Soulier et al., 2021b). Dedicated measurements should be performed for further investigations of this delay.

**Technical corrections** The technical corrections have been treated directly in the manuscript

**References**

[revised manuscript text omitted]

---

## Referee Report (RR1)

**General Comments**

The topic of the manuscript for review, namely the characteristics of a novel device for the detection of flow separation and the onset of stall, is of relevance for the wind energy community. Different sensor setups and locations have been investigated in wind tunnel experiments at Re=885.000 for this purpose. The characterization of the sensor performance dependency on angle of attack is carried out mainly by comparison of the mean signal and its standard deviation. The presented results indicate the ability of the novel sensor to identify the onset of trailing edge separation and the stall angle. This ability is reduced in some configurations due to the position of the sensor.
The presented experimental investgations provide interesting information about the sensor characteristics and possible limitations for certain use cases, depending on sensor location. However, the description of the results could be improved by taking the detected pressure distribtions into account in order to explain the sensor sensitivity differences at certain positions of the airfoil. These improvements can be easily done based on the already existing plots.

I recommend this paper for publication after very minor revisions.

**Specific Comments**

- In section 3 (p. 8, line 104f) the authors mention, that transition from an attached to seprated flow can be seen "from the intermittent appearance of a plateau on the pressure distribution". It is unclear, how the authors define this "intermittency", since only temporally and spatially averaged pressure distributions for two AoAs (6° and 8°) are plotted. One could argue, that for intermediade AoAs a gradual increase of the plateau is expected, rather than an intermitted occurence.

- In section 3 (p. 9, line 111f) the authors mention, that the flow separation location is "really close to the leading edge". An estimate in terms of chordwise location should be given based on the information from the pressure distributions.

- On page 10, figure 10 is presented without any reference in the text. The only reference to figure 10 appears on page 14. The figure should be moved closer to the reference in the text or a reference / explanation in the text should be given close to its current position.

- In section 4.1 (p.11, lines 137ff) the authors describe the increase of the sensor signal in the AoA range of 5° to 8°, which is followed by decreasing "linearly" to 0. On the one hand, linearity is hard to conlcude from just three measurement points. On the other hand, the authors do not give any explanation or hypothesis for this unexpected behaviour. At least an attempt to reason this sudden reduction in signal value should be undertaken.

- In section 4.1 (p.11, lines 140) the authors claim that the sensor is appropriate for the detection of TE flow separtion "at least for this type of profile". Which limitations on the usability would the authors expect, and

for which other types of profile? What would be the benefit of a sensor, that can only be appropriately used for one type of profile? Although I don't expect significant limitations for typical wind energy profiles, the authors should give an explanation or argument once they raise this concern.

- In section 4.3 the authors compare the signals of the sensor depending on loaction close to the TE or LE. Figure 13 (a and b) show strong increases of signal and standard deviation for the LE loaction once the AoA reaches the stall region. It would be of great benefit for the reader to relate the position of the sensor to the actual position of the flow separation as it can be concluded from the pressure distributions. An assumption would be, that the LENSL sensor is located upstream of the location of flow separation until AoA 18°. A comparison of sensor location and separation location would help to understand the reason for the low responsiveness in lower AoAs and the sudden signal increase afterwards, though.

**Technical Comments**

The labeling of the plots in figure 11 to 14 lacks a proper label for the y-axes as only the unit [Volts] are given. Meaningful labels should be added

The formatting of physical units, mathematical symbols and abbreviations should be checked and modified in compliance with the guidelines (upright vs. italics; blank spaces) .
`https://www.wind-energy-science.net/submission.html#math`

p.1, line 21:
Year of publication is missing.

p.2, line 33:
Format of Reynolds number: Use `\cdot` or `\times` instead of x or blank space. Several different instances occur during the course of the manuscript.

p.2, line 43:
Subsection: consitent use of uppercase and lowercase initials.

p.3, line 54:
stalled regulated -> stall-regulated

p.10, figure 9:
There is no left plot in the manuscript draft, just two plots underneath each other. A consitent label a) and b) would be helpful.
The placement of the legend in the zoomed-in figure should be changed so no plot is covered by it.

p.13, figure 12 b, p.14, figure 13 b:
Legend "mean TENS* signal" should be "Standard deviation"

p.13, line 182:
there -> they

p.15, line 204:
shorten -> shortened

p.15, line 206:
have -> has

p.6, line 212:
have -> has

---

## Author Response (AR2)

**Response to referee #1**

**General Comments**

The topic of the manuscript for review, namely the characteristics of a novel device for the detection of flow separation and the onset of stall, is of relevance for the wind energy community. Different sensor setups and locations have been investigated in wind tunnel experiments at Re=885.000 for this purpose. The characterization of the sensor performance dependency on angle of attack is car- ried out mainly by comparison of the mean signal and its standard deviation. The presented results indicate the ability of the novel sensor to identify the onset of trailing edge separation and the stall angle. This ability is reduced in some configurations due to the position of the sensor. The presented experimental investigations provide interesting information about the sensor characteristics and possible limitations for certain use cases, depend- ing on sensor location. However, the description of the results could be improved by taking the detected pressure distribution into account in order to explain the sensor sensitivity differences at certain positions of the airfoil. These improve- ments can be easily done based on the already existing plots. I recommend this paper for publication after very minor
**Response:** We would like to thank the referee for taking the time to evaluate our manuscript. We took his comments into account which improved the quality of the manuscript.

**Specific Comments**

**Referee:** In section 3 (p. 8, line 104f) the authors mention, that transition from an attached to separated flow can be seen "from the intermittent appearance of a plateau on the pressure distribution". It is unclear, how the authors define this "intermittency", since only temporally and spatially averaged pressure distributions for two AoAs (6° and 8° ) are plotted. One could argue, that for intermediade AoAs a gradual increase of the plateau is expected, rather than an intermitted occurence.
**Response:** This is true that we only provided the average value of the pressure distribution. For consistency with the presented results, the sentence has been replaced by:
"Between $AoA \simeq 6\,°$ to $8\,°$, the flow is transitioning from the attached state to the separated state (i.e. zero pressure gradient), as can be seen on the zoom of figure 9, where the mean pressure coefficient near the trailing edge is progressively increasing towards a plateau from $AoA = 6\,°$"

**Referee:** In section 3 (p. 9, line 111f) the authors mention, that the flow separation location is "really close to the leading edge". An estimate in terms of chordwise location should be given based on the information from the pressure distributions.
**Response:** a chordwise location has been added.

**Referee:** On page 10, figure 10 is presented without any reference in the text. The only reference to figure 10 appears on page 14. The figure should be moved closer to the reference in the text or a reference / explanation in the text should be given close to its current position.
**Response:** Thank you for noticing. This figure is not comparing the TESL case with the reference case, it has thus no meaning in page 14 and the related sentence has been removed from the paragraph. With no reference in the document, the figure has been removed from the article.

**Referee:** In section 4.1 (p.11, lines 137ff) the authors describe the increase of the sensor signal in the AoA range of 5 ° to 8 ° , which is followed by decreasing "linearly" to 0. On the one hand, linearity is hard to conlcude from just three measurement points. On the other hand, the authors do not give any explanation or hypothesis for this unexpected behaviour. At least an attempt to reason this sudden reduction in signal value should be undertaken.
**Response:** Thank you for these remarks that were taken in to account together with a possible scenario

of what is happening as follows:

"From -5 ° to 5 °, the linear evolution of the lift indicates an attached flow state. For this *AoA* range, both the mean and the standard deviation of the signal of the e-Telltale are near 0. From 5 ° to 8 °, corresponding to the flow separation appearance at the trailing edge, both the mean and the standard deviation rise up with the *AoA*. Then from 8° to 10° they decrease to values near 0. It should be noticed that this behaviour is not present when the e-Telltale is located near the leading edge (LENSL, figure **??**), clearly indicating that this bump from 5° to 10° is associated to the location of the e-Telltale sensor in the trailing edge flow separation area. This suggests that, in that region, when the AOA is increasing from 5 ° to 8 °, the flow is departing from the wall causing movement of the e-TellTale strip outward the wall, thus increasing its signal. The separated flow should naturally induces a mean shear area with associated turbulent structures. This turbulent separated shear layer is supposedly fluctuating from the wall to the separated shear area, causing fluctuations of the e-Telltale strip (that was observed), increasing the RMS value of the e-Telltale sensor. Above 8 °, the separated shear layer is probably moving to far from the wall, getting out-of-reach of the e-TellTale strip length, which in turn reduces the mean and RMS value of the e-TellTale signal. More spatio-temporal informations are needed to confirm this scenario. Also, it should be noted that this scenario do not take into account potential fluid-structure interactions with the e-TellTale strip."

**Referee:** In section 4.1 (p.11, lines 140) the authors claim that the sensor is appropriate for the detection of TE flow separtion "at least for this type of profile". Which limitations on the usability would the authors expect, and for which other types of profile? What would be the benefit of a sensor, that can only be appropriately used for one type of profile? Although I don't expect significant limitations for typical wind energy profiles, the authors should give an explanation or argument once they raise this concern.

**Response:** For the blade shape tested, the flow separation starts at the trailing edge and then move progressively towards the leading edge with the increase of the angle of attack. The two states (trailing-edge separation and leading-edge separation) are clearly separated, and most of wind turbine blade shapes behave similarly. However, for thin airfoils used in aircraft applications for instance, a strong stall occurs just after the maximum lift value, with a very fast displacement of the separation from the trailing-edge to the leading edge. In that case the dynamic response of the E-TellTale sensor is of major importance. Also, the distance of the separated shear layer to the e-TellTale strip might be too far for a sensor located at the trailing edge.

This wasn't the good location to put some limitations the sensor. The sentence "at least for this type of profile" has been remove and we can now read in the conclusion:

"At last, the sensor might have some limitations when dealing with thin airfoils that have a strong stall just after the maximum lift value, with a very fast displacement of the flow separation from the trailing-edge to leading-edge. In that case the dynamic response of the sensor is of major importance and the best location might be rather near the leading-edge. Indeed, the sensitivity of the sensor to the separated shear layer when it is located at the trailing edge may rapidly disapear."

**Referee:** In section 4.3 the authors compare the signals of the sensor depending on location close to the TE or LE. Figure 13 (a and b) show strong increases of signal and standard deviation for the LE location once the AoA reaches the stall region. It would be of great benefit for the reader to relate the position of the sensor to the actual position of the flow separation as it can be concluded from the pressure distributions. An assumption would be, that the LENSL sensor is located upstream of the location of flow separation until AoA 18° . A comparison of sensor location and separation location would help to understand the reason for the low responsiveness in lower AoAs and the sudden signal increase afterwards, though.

**Response:** This is indeed the case. The link with the separation area is now clearly stated:

"This should be attributed to the separation location, where the zero mean pressure gradient starts, that is reaching the strip region, between 19 % to 31 % of the chord length, for angles of attack higher

than 18 °.''
**Technical Comments**

**Response:** Thank you for these corrections. All the technical issues were addressed.

**Response to referee #2**

**Response:** Thank you for these corrections. All the technical issues have been addressed.